# Assessment of Intercropping and Plastic Mulch as Tools to Manage Heat Stress, Productivity and Quality of Jalapeño Pepper

**Jesús Santillano-Cázares [1]** **, Cristina Ruiz-Alvarado [1], Alejandro M. García-López [1], Isabel Escobosa-García [1], Víctor Cárdenas-Salazar [1], Antonio Morales-Maza [2] and Fidel Núñez-Ramírez [1],***

[1] Instituto de Ciencias Agrícolas, Universidad Autónoma de Baja California, Carretera a Delta s/n Ejido Nuevo León, Baja California C.P. 21705, Mexico; jsantillano@uabc.edu.mx (J.S.-C.); mariacristina@uabc.edu.mx (C.R.-A.); amgarcial@uabc.edu.mx (A.M.G.-L.); isabel.escobosa@uabc.edu.mx (I.E.-G.); victorcardenas@uabc.edu.mx (V.C.-S.)

[2] Instituto Nacional de Investigaciones Forestales, Agrícolas y Pecuarias (INIFAP), Campo Experimental Valle de Mexicali, Mexicali, Baja California C.P. 21000, Mexico; morales.antonio@inifap.gob.mx

* Correspondence: fidel.nunez@uabc.edu.mx; Tel.: +52-686-523-0079

**Abstract:** Under a global warming scenario, it is important to adopt practices that favor soil water conservation, such as plant intercropping systems and the use of plastic mulching. The objective of this study was to determine how microenvironment, morphology, productivity and quality of jalapeño peppers were affected by corn intercropping and the use of plastic mulching. Two experiments were conducted during 2015 and 2016 in the Valley of Mexicali, Mexico, a region characterized by its extreme aridity, soil salinity, hot temperatures and high radiation during the summer. Four treatments were tested: jalapeño peppers grown on bare soil (BS); on bare soil intercropped with corn (BS+IC); on plastic mulch (PMu); and on plastic mulch intercropped with corn (PMu+IC). The response variables measured were yield, fruit quality attributes, microclimatic variables, and morphology of the pepper crop. PMu treatment produced the tallest pepper plants and yields, while the BS+IC treatment produced the smallest plants and the lowest yields. A possible explanation for the higher biomass and crop yield of the PMu treatment is the lack of competition from corn and the effect of plastic mulching in reducing soil salinity. It is concluded that competition from corn on jalapeño pepper dramatically affected the pepper's productivity, particularly under high soil salinity and extremely high temperature conditions.

**Keywords:** global warming; climate change; soil salinity; shading; corn; plant competence

## 1. Introduction

The southwest of the United States and northwest Mexico are among the most extreme climatic regions in the world, and models predict that climatic conditions will become even hotter and drier in the future [1,2]. Among the most concerning consequences of global warming are the negative effects on crop production, key for human subsistence [3–5]. Lobell and Ortiz-Monasterio (2007) [6] predicted that an increase of 1 °C in maximum temperatures in the valley of Mexicali (in Baja California, Mexico) would have a greater effect on wheat (*Triticum aestivum* L.) yields (−5.1%) than an increase in minimum temperatures (−1.8%). These model-based predictions seem to have become reality, and wheat yields in the region have decreased from 6.5 t ha$^{-1}$ produced until one decade ago, to less than 6.0 t ha$^{-1}$ produced during the last five to six years. In relation to global warming and related heath stress in plants, López-Marín et al. (2011) [7] pointed out that heat and light stresses inflicted a physiological

disorder on various crops grown in warm or semi-arid climates in Spain and proposed shading as a means to alleviate these effects. In this regard, intercropping systems can provide shading and thus create a more suitable microenvironment for the shorter crop´s canopy [5].

Intercropping is defined as a strategy of growing two or more crops in the same field at the same time [8]. Intercropping has been long known to have several advantages, such as land use efficiency, extended cropping season and longer soil protection from sun, rain and weeds. This system provides an opportunity for a greater variety of crops that can be grown and to produce higher yields when the system´s productivity is assessed as a whole [9]. More recently, intercropping has been proposed as a tool to adapt to climate change [10,11] and as a system that favors sustainability [12,13]. However, intercropping does not always have positive effects on one or both crops grown, since the effect of shading within the intercropping systems can have varying effects on the yield of the shaded crop. For example, morphological changes in soybeans due to shading include changes in internode length, plant height, leaf size, and branching [5,14,15]. Ramamurthy et al. (1993) [16] found that intercropping pepper with finger millet (*Eleusine coracana*) reduced pepper yields by 88% compared with pepper grown as monocrop. It was suggested that shading of the pepper plants by the finger millet was an important cause for the pepper´s yield reduction. Similarly, a negative effect of shading by intercropped corn on the yield of soybeans (*Glycine max*) was observed by Liu et al. (2017) [17]. It should be noted that in the present study, intercropping was examined as a tool to alleviate heath stress of the shorter crop (the crop of interest), instead of for its documented ability to increase land use efficiency.

On the other hand, while Duchene et al. (2017) [12] recognized that competition was almost unavoidable within plant species grown in intercropped systems, this competition is characterized as a complex interaction that would not necessarily hurt the species in system. To explain this effect, they suggested that the species that is being shaded could suffer from lower quantity and quality of the incident light (as pointed by Liu et al., 2017 [17]) but, at the same time, could benefit from cooler canopy and soil temperatures, increased soil microbial activity, water availability, and nutrient uptake. The final result of intercropping depends on the morphological and physiological adaptations of the species in the system. Duchene et al. (2017) [12] supports early conclusions by Mead and Riley (1981) [9] in that intercropping systems can produce very different results in different regions, due to the complex interactions caused by differences in the species selection, particular ecological conditions and prevailing management practices.

In a global warming scenario, it is important to adopt practices that favor water conservation for transpiration, which is a key mechanism for plant cooling and wellbeing. Plastic mulches are used in agriculture as a tool for water conservation [18–20]. Zhang et al. (2017b) [21] found that topsoil temperature was significantly ($p < 0.05$) higher in field plots with plastic film mulch than the control, and resulted in greater soil water storage up to 40 days after planting. Díaz-Pérez (2010) [22] found that microenvironment, physiology and yields of bell peppers were influenced by different color plastic mulches. It is of interest to examine the combined effects of intercropping and mulching in northwest Mexico´s highly productive crop production systems, which are characterized by intensive input use, but with low use efficiency [23]. Intercropping systems are not commonly utilized in northwest Mexico, one region where the effects of global warming in crop production are already having a negative impact. The objective of this study was to assess how microenvironment, morphology, productivity and quality of jalapeño peppers are affected by intercropping with corn and plastic mulching.

## 2. Materials and Methods

### 2.1. Site Description

Two experiments were conducted during the spring of 2015 and 2016 at the experimental field of the Instituto de Ciencias Agrícolas, Universidad Autónoma de Baja California (ICA-UABC), at Ejido Nuevo León, Mexicali, B.C., Mexico (lat. 32° 24′ 26.38″ N, long. 115° 11′ 55.43″ W, elevation 14 m a.s.l.).

The region is characterized by desertic very arid, with extremes ranging over 14 ° C (BW (e′)) [24]. The mean annual temperature is 22.3 °C, but can range from as high as 50 °C during the summer, to as low as −7 °C during the winter. Mean annual precipitation is only 58 mm. The soil is classified as Aquic haplotorrert Smectitic, Hyperthermic, Haplic Vertisol (calcaric, Endogleyic), with a pH of 7.8.

*2.2. Crop Management*

Two irrigation polyethylene drip tapes with drippers (1 L h$^{-1}$ per emitter), placed every 40 cm were laid on raised beds of 0.8 m width with beds spaced at 1.6 m. Jalapeño peppers (*Capsicum annuum* L.) (Colima ACX 117) (Abbott & Cobb, Inc.) were transplanted on 10 and 11 March 2015 and 2016, respectively. Transplanting was made on double rows per bed, with 40 cm spacing between plants within the same row. Corn cultivar XR66 (Ceres, Semillas de Mexico) was used for intercropping in both years. In both years, the fertilization for all treatments consisted of 250-85-100 kg ha$^{-1}$ of nitrogen (N), phosphorus (P$_2$O$_5$), and potassium (K$_2$O), using urea (46-00-00), ammonium sulfate (21-00-00), monoammonium phosphate (11-52-00), and potassium nitrate (13-00-46) as commercial fertilizer sources. The fertilization was applied weekly through drip irrigation and was equally split during the growing season of the peppers. Irrigation events were applied differentiating among mulched and non-mulched treatments based on soil water availability, when soil water tension reached 25 kPa, as indicated by tensiometers that were placed at a depth of 30 cm in the soil profile. The tensiometers were distributed across the entire experimental area. During the study, pests *Diabrotica* spp. and *Mysus persicae* (Sulzer) were present and controlled with Bifentrin (18.48%) + Imidacloprid (53.70%) and zeta-cypermethrin + bifenthrin, using recommended doses.

*2.3. Treatments Description*

Four treatments were tested: jalapeño peppers grown on bare soil (BS), jalapeño peppers grown on bare soil intercropped with corn (BS+IC), jalapeño peppers grown on plastic mulch (PMu), and jalapeño peppers grown on plastic mulch intercropped with corn (PMu+IC). In PMu and PMu+IC treatments, white plastic mulch was used in 2015 and transparent plastic mulch in 2016. For intercropped treatments (BS+IC and PMu+IC), corn was planted the same day the peppers were transplanted. In treatments with intercropping, the corn plants were alternated with pepper plants within rows, i.e., within each row there was a corn plant between each pepper plant. Pepper plant density per unit area was the same in all four treatments (a mean of 3.12 plants per m$^2$). Because of the switch of plastic mulch colors from one year to the other, data was analyzed for each of the two years independently. The treatment design in both years was a completely randomized design with four replications.

*2.4. Response Variables*

The response variables measured in the study were the pepper yield, fruit number, pepper quality (capsaicin concentration, pericarp wall thickness, firmness of the fruits, and color attributes (L*, a*, b*, Chroma and Hue)), microclimatic variables affecting the peppers (photosynthetically active radiation (PAR), temperature, and relative humidity), and morphology (height of the plants, leaf area (LA), specific leaf area (SLA), leaves dry weight (LDW), stems dry weight (SDW), fruit dry weight (FDW), and total plant dry weight (TDW)). When reached physiological maturity (green stage), fruit peppers were harvested at 72, 79, and 87 days after transplanting (DAT) in 2015 (on 21 and 28 May, and 5 June) and at 75, 85, and 92 DAT in 2016 (on 25 May, and 4 and 11 June). At each of these harvest dates, yield and number of fruits were estimated, and four fruits were randomly selected to determine chemical and quality attributes. The capsaicin concentration was measured according to Davis et al. (2007) [25]. The pericarp wall thickness was measured with a digital Vernier. The firmness of the fruits was measured in Newtons (N) using a texturometer Chatillon (AMETEK, Inc., Berwyn, PA, USA). Fruit skin color attributes (L*, a*, b*, Chroma and Hue) were measured with spectrophotometry, using a X-Rite SP-62 model (Metrolab Internacional, Co. Monterrey Mexico). Photosynthetic active radiation (PAR) was measured on 12 May and 2 June 2015, and on 27 April and 13 May 2016, using a Watch Dog sensor

(Spectrum Technologies' LightScout, IL, USA) which was located 20 cm above the pepper canopy. The temperature and relative humidity were measured at the pepper canopy height on 16 April and 12 May 2016, using a digital thermohygrometer CEM DT-172 (Twin Light Instruments, Monterrey, Mexico). Plant height was measured 8 times during the season in both years. Final measurements occurred at 86 and 90 DAT, in 2015 and 2016, respectively. The LA was measured with a leaf area meter (LI-3100 Area Meter, NE, USA), and the SLA was measured according to Hunt (2002) [26]. To determine LDW, SDW, and FDW, the different plant parts (leaves, stems and fruits) were dried in a forced-air drying oven at 60 °C for 48 h. All the morphology variables were measured 3 times per year in both years, but only the final values measured 98 DAT in both years are reported.

## 2.5. Experimental Design and Analyses

Data were analyzed by individual years (due to switching mulch colors from year to year), using MINITB14®. Although all response variables were measured varying number of times during the growing cycle in both years, since no consistent effects were found by dates, yields and fruit number are reported as the sum of all sampling dates; pepper quality attributes are reported as the mean of all sampling dates; and pepper plant morphology characteristics are reported only from the final date of measurements. When statistical differences were detected by analyses of variance (ANOVA), the Tukey test was used to compare differences among treatment means. The probability level to declare differences among treatments means was equal or lower than 0.05.

# 3. Results

## 3.1. Yield and Fruit Number

Pepper yields were significantly ($p < 0.05$) affected by the treatments in both years (Table 1). The BS and PMu treatments did not significantly differ from each other ($p < 0.05$) and were higher than the treatments with intercropping, which were not significantly different from each other ($p > 0.05$). Across years, mean yields were 11.3, 5.1, 13.0, and 4.6 Mg ha$^{-1}$ for treatments BS, BS+IC, PMu, and PMu+IC, respectively. Treatments without intercropping had yields that were 2.5 times higher than the treatments with intercropping. The fruit number in both years was significantly ($p < 0.05$) different among treatments (Table 1), with the treatments without intercropping having a higher number of fruits.

**Table 1.** Jalapeño pepper yields, fruit number, capsaicin concentration, pericarp thickness and fruit firmness as affected by plastic mulch and intercropping, during two growing seasons.

| Trt. [†] | Total Yield (Mg ha$^{-1}$) | | Total Fruit Number (m$^2$) | | Capsaicin (g kg FW [¥]) | Pericarp Thickness (mm) | Firmness (N) | |
|---|---|---|---|---|---|---|---|---|
| | | | Season 2015 | | | | | |
| BS | 10.7 | a [‡] | 56.3 | a | 1.1 | 3.6 | 75.4 | |
| BS + IC | 4.6 | b | 21.8 | b | 0.9 | 3.4 | 78.0 | |
| PMu | 12.1 | a | 62.3 | a | 1.0 | 3.4 | 67.7 | |
| PMu + IC | 4.4 | b | 15.7 | b | 1.0 | 3.4 | 81.2 | |
| *p*-value | 0.005 | | 0.007 | | 0.431 | 0.422 | 0.331 | |
| | | | Season 2016 | | | | | |
| BS | 11.8 | a | 65.6 | a | | 3.0 | 63.8 | a |
| BS + IC | 5.6 | b | 28.4 | b | | 3.3 | 36.6 | c |
| PMu | 13.9 | a | 73.4 | a | | 3.4 | 50.3 | a |
| PMu + IC | 4.7 | b | 33.7 | b | | 3.1 | 47.4 | b |
| *p*-value | 0.001 | | 0.001 | | | 0.788 | 0.028 | |

[†] Peppers grown on bare soil (BS), on bare soil intercropped with corn (BS+IC), on plastic mulch (PMu), and on plastic mulch intercropped with corn (PMu+IC); [‡] numbers followed by different letters within a column are significantly different among treatments (Tukey: $p < 0.05$); [¥] Fresh weight.

## 3.2. Pepper Quality Variables

Capsaicin concentration was determined only in 2015 and was not significantly ($p > 0.05$) different across treatments (Table 1). In 2015 or 2016, the pericarp wall thickness was not significantly different across treatments ($p > 0.05$) (Table 1). Fruit firmness was significantly different ($p < 0.05$) only in 2016, with non-intercropping treatments being higher than treatments with intercropping. Fruit color attributes L\*, a\*, b\*, Chroma, or Hue were not affected ($p > 0.05$) by the treatments in either year (Table 2).

**Table 2.** Jalapeño pepper color variables (L\*, a\*, b\*, Chroma, and Hue) as affected by mulching and intercropping, during two growing seasons.

| Trt. [†] | L\* | a\* | b\* | Chroma | Hue |
|---|---|---|---|---|---|
| Season 2015 | | | | | |
| BS | 36.4 | −9.4 | 18.7 | 21.0 | 116.8 |
| BS+IC | 38.5 | −9.8 | 22.6 | 24.8 | 114.7 |
| PMu | 37.6 | −9.9 | 20.7 | 23.0 | 116.3 |
| PMu+IC | 37.0 | −9.9 | 20.9 | 26.6 | 116.0 |
| *p*-value | 0.67 | 0.47 | 0.328 | 0.431 | 0.47 |
| Season 2016 | | | | | |
| BS† | 34.8 | −9.5 | 21.6 | 23.1 | 115.9 |
| BS+IC | 36.3 | −10.0 | 21.2 | 24.5 | 116.2 |
| PMu | 35.9 | −9.7 | 19.5 | 23.3 | 117.5 |
| PMu+IC | 35.4 | −10.3 | 22.6 | 24.2 | 114.6 |
| *p*-value | 0.604 | 0.385 | 0.709 | 0.671 | 0.704 |

[†] Peppers grown on bare soil (BS), on bare soil intercropped with corn (BS+IC), on plastic mulch (PMu), and on plastic mulch intercropped with corn (PMu+IC).

## 3.3. Microclimatic Variables on the Pepper Crop

Intercropping of corn in pepper caused microclimatic conditions in the jalapeño pepper plants' environment. The PAR measurements collected on 12 May and 2 June 2015, and 27 April and 13 May 2016, consistently showed a PAR reduction in pepper plants that grew under intercropping, compared with those that grew without intercropping (under full sunlight) (Figure 1). In the two sampling dates of 2015, PAR on the intercropped treatments (from 07:00 to 19:00) averaged 151 $\mu mol\ m^{-2}\ s^{-1}$ and 380 $\mu mol\ m^{-2}\ s^{-1}$, compared with 787 $\mu mol\ m^{-2}\ s^{-1}$ and 1162 $\mu mol\ m^{-2}\ s^{-1}$, on full sunlight pepper treatments, respectively. In 2016, corresponding PAR mean values were of 360 $\mu mol\ m^{-2}\ s^{-1}$ and 266 $\mu mol\ m^{-2}\ s^{-1}$, on the shaded pepper plants, compared with 916 $\mu mol\ m^{-2}\ s^{-1}$ and 1147 $\mu mol\ m^{-2}\ s^{-1}$, on full sunlight pepper plants for the two sampling dates, respectively. Temperature at the pepper canopy level also decreased in intercropped treatments compared with treatments without intercropping (Figure 2). Maximum temperatures at noon recorded on 16 April and 12 May 2016 were 62 and 65.5 °C, respectively, for pepper as monocrop, while temperatures at the same hour in the pepper canopy under intercrop were 34.9 and 37.3 °C, representing differences of 27.1 and 28.2 °C, respectively.

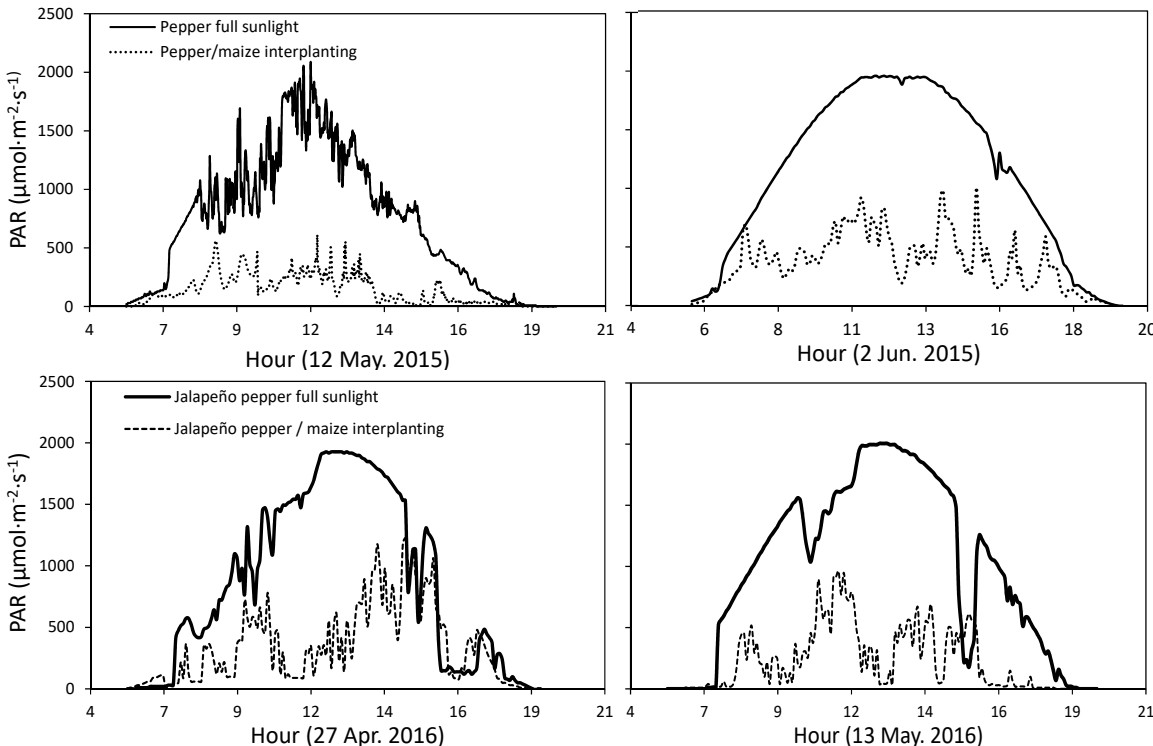

**Figure 1.** Photosynthetic active radiation (PAR) distribution of a jalapeño pepper canopy as affected by intercropping, during 2015 and 2016.

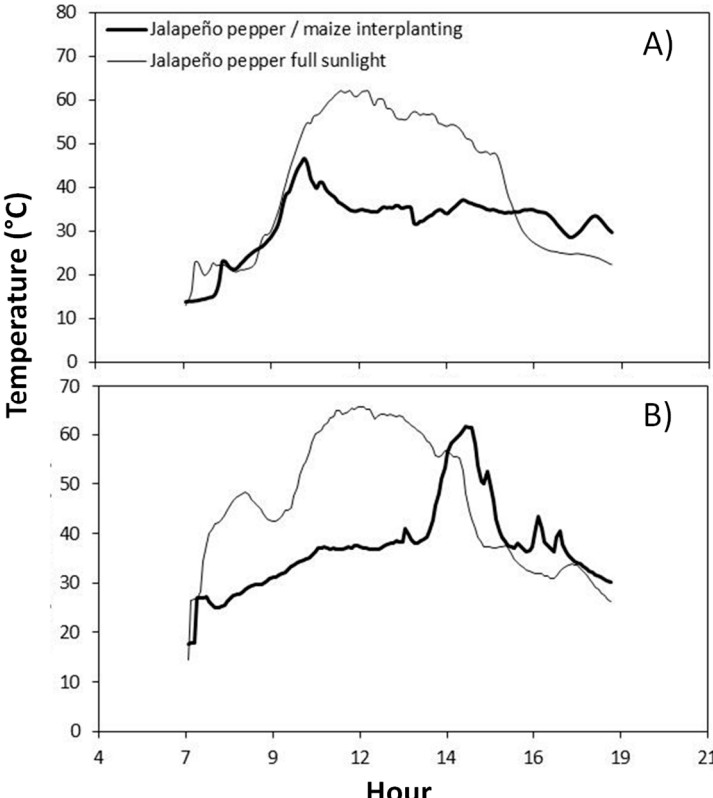

**Figure 2.** Temperatures recorded at a jalapeño pepper canopy level as affected by intercropping, on (**A**) 16 April and (**B**) 12 May 2016.

### 3.4. Pepper Plant Morphology

In 2015, pepper plant height was significantly ($p > 0.05$) affected by treatments (Table 3). The treatment PMu produced significantly ($p < 0.05$) taller pepper plants than the rest of the treatments. In 2016, plant height was again significantly ($p < 0.05$) affected by the treatments (Table 3). BS and PMu+IC produced shorter pepper plants than BS+IC or PMu, which performed similarly ($p > 0.05$) and produced the tallest pepper plants. In 2015, treatments significantly ($p < 0.05$) affected LA, with PMu treatment having higher LA than the rest of the treatments. The opposite was recorded for treatment BS+IC, which recorded the lowest LA. In 2016 again, treatments were significantly different ($p < 0.05$) among treatments. Once again, PMu had substantially higher LA than the rest of the treatments and BS+IC the lowest. The difference between PMu and BS+IC was 6.2-fold. SLA was not significantly ($p > 0.05$) different in either of the years. Mean LDW both in 2015 and 2016 was significantly ($p < 0.05$) different among treatments, with PMu treatment recording heavier leaves than the rest of the treatments (Table 3). In contrast, BS+IC consistently recorded the lightest leaves, while BS and PMu+IC were intermediate. The mean SDW was significantly ($p < 0.05$) different among treatments in both years, with PMu treatment recording the heaviest stems than any of the other treatments, while the opposite was recorded for treatment BS+IC, which had the lightest stems (Table 3). The mean FDW was different ($p < 0.05$) among treatments in both years (Table 3). Just as observed with most other plant morphology variables, treatment PMu produced the heaviest fruits and BS+IC the lightest. Consequently, the TDW was significantly ($p < 0.05$) different among treatment means in both years (Table 3). The recorded TDW was much higher for PMu treatment than any other treatment.

**Table 3.** Jalapeño pepper height, leaf area (LA), specific leaf area (SLA), leaves dry weight (LDW), stems dry weight (SDW), fruit dry weight (FDW), and total dry weight (TDW) as affected by mulching and intercropping, during two growing seasons.

| Treatment [†] | Height (cm) | | LA (cm$^2$) | | SLA (cm$^2$ g$^{-1}$) | LDW (g) | | SDW (g) | | FDW (g) | | TDW (g) | |
|---|---|---|---|---|---|---|---|---|---|---|---|---|---|
| | | | | | Season 2015 | | | | | | | | |
| BS | 41.7 | b [‡] | 1229.0 | b | 100.3 | 12.3 | b | 16.5 | ab | 26.3 | ab | 55.0 | b |
| BS+IC | 37.1 | b | 470.8 | b | 99.1 | 4.8 | b | 6.0 | b | 9.0 | b | 19.8 | b |
| PMu | 52.9 | a | 4094.7 | a | 123.2 | 33.3 | a | 24.5 | a | 67.3 | a | 125.0 | a |
| PMu+IC | 40.9 | b | 1773.0 | b | 169.8 | 10.5 | b | 10.8 | ab | 25.3 | ab | 46.5 | b |
| *p*-value | <0.001 | | 0.009 | | 0.352 | 0.006 | | 0.032 | | 0.048 | | 0.004 | |
| | | | | | Season 2016 | | | | | | | | |
| BS | 53.0 | b | 1128.2 | bc | 92.2 | 12.2 | b | 17.5 | ab | 28.6 | b | 58.3 | b |
| BS+IC | 65.4 | a | 619.8 | c | 111.6 | 5.6 | c | 7.6 | c | 12.7 | c | 25.8 | c |
| PMu | 63.1 | a | 3854.4 | a | 136.7 | 28.2 | a | 25.8 | a | 67.2 | a | 121.2 | a |
| PMu+IC | 51.9 | b | 1694.6 | b | 116.3 | 14.6 | b | 12.2 | b | 29.2 | b | 56.1 | b |
| *p*-value | <0.001 | | 0.004 | | 0.457 | 0.018 | | 0.039 | | 0.027 | | 0.002 | |

[†] Peppers grown on bare soil (BS), on bare soil intercropped with corn (BS+IC), on plastic mulch (PMu), and on plastic mulch intercropped with corn (PMu+IC); [‡] numbers followed by different letters within a column are significantly different (Tukey: *p* < 0.05).

## 4. Discussion

Climatic conditions occurring during both years indicate that the pepper plants' physiology and yields, quality and morphology were probably affected, since temperatures over 40 °C occurred during the last 30 days of the peppers' growing season in both years (Figure 3). Early reports [27] indicate that air temperature thresholds for causing sunburn injury in cucumbers and peppers are 38 to 40 °C and 40.5 to 42.5 °C, respectively. These authors suggested that sunburn occurs when photosynthesis is disturbed by excessive heat, so that light energy is redirected into damaging light-dependent processes. In 2015, 1575 growing degree days were recorded, while in 2016, 1515 growing degree days occurred.

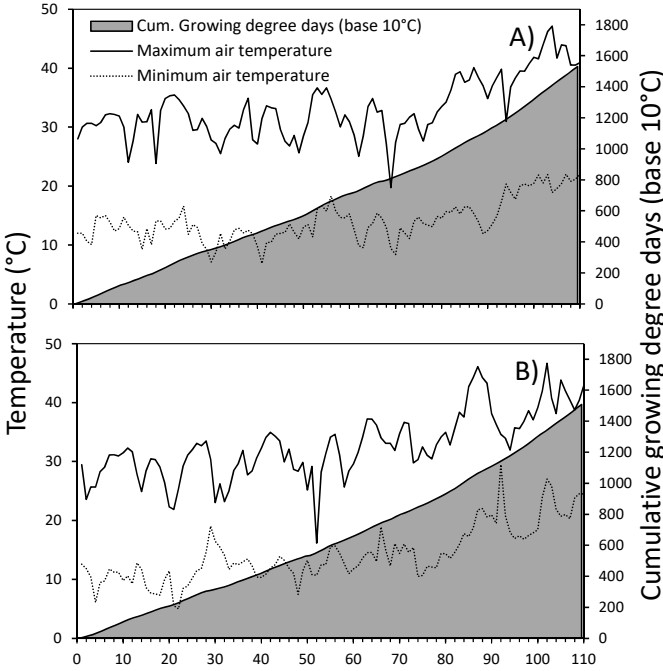

**Figure 3.** Actual maximum and minimum daily air temperatures with cumulative growing degree days, during experiments in 2015 (**A**) and 2016 (**B**).

### 4.1. Yield and Fruit Number

Ghanbari et al. (2010) [28] concluded that intercropping in Iran achieved the highest light interception, soil moisture, soil temperature and yield, compared to sole crops. Furthermore, microclimatic variation exerted by intercropping caused favorable environmental conditions, conducive for growth and high yield compared to sole crops. Although the intercropped corn provided shade for the pepper plants (Figure 2), protecting the plants from stress associated with excessive solar radiation and temperature and, in theory, supporting yields, this did not occur in our study possibly due to an overriding effect of competition for light, and possibly nutrients, water, space, and other resources. Yang et al. (2017) [29] reported that in a corn–soybean intercropping system, the corn aggressively competed with soybeans and depressed the soybeans yields. Díaz-Pérez (2013) [30] concluded that high shading levels (by shading nets) on bell peppers reduced leaf temperature and excessive transpiration, but resulted in reduced leaf photosynthesis. Similarly, Ramamurthy et al. (1993) [16] concluded that intercropping pepper with the cereal finger millet (*Eleusine coracana* L. Gaertn) reduced pepper yield by as much as 88%, compared with pepper as monocrop. It was hypothesized that shading of the pepper plants by the finger millet was a factor in pepper yield reduction. Liu et al. (2017) [17] reported that positive morphological changes of soybeans growing in intercropping with corn were not able to compensate for the effect of reduced leaf area (due to smaller leaf size and less branching) and total light interception, leading to reduced soybeans biomass and grain production.

Among competition-free treatments (BS and PMu), consistent with previous research carried out on potato [31], the application of plastic mulch increased the yield in PMu in both years. Plastic mulching is reported to reduce soil salinity [32–34]. A soil analysis of the experimental area from the previous growing season showed an electric conductivity of 5.44 dS/m. Aragüés et al. (2014) [32] found that soil electrical conductivity, chloride and sodium adsorption ratio values were much lower in mulched treatments than in bare soil treatments due to reduced evaporation losses and related decreases in salt concentrations. In Iran, Sedaghati et al. (2016) [33] reported that the use of plastic mulch decreased the soil surface salinity, compared to treatments without plastic mulching. Similarly in China, Zhao et al. (2016) [34] found that soil salinity was always higher under bare ground than under plastic mulch. In this study, the competition caused by corn intercropping in addition to the likely soil salinity effect in non-mulched treatments represented strong limiting factors for pepper production. While our mean yields ranged across years from 4.5 to 13 t ha$^{-1}$ (PMu+IC and PMu, respectively), the mean yields in coastal municipalities of Baja California are reported to be 25.6 t ha$^{-1}$ [35]. The mean jalapeño pepper yields for the same year (2013) for one of the most emblematic Mexican jalapeño producers, Chihuahua, was 33.2 t ha$^{-1}$ [35]. Inzunza-Ibarra et al. (2007) [36] reported jalapeño pepper yields in Durango ranging from 28.9 to 56.4 t ha$^{-1}$. Thus, as observed by the large gap between our yields and other comparable reports, jalapeño peppers in our study evidently were under severe stress conditions imposed by high temperatures occurring toward the end of the experiment (Figure 3), the competition caused by intercropping, by soil salinity, and their concomitant negative effects.

### 4.2. Pepper Quality Variables

Although no statistical differences ($p > 0.05$) were detected among treatments, in general, absolute capsaicin concentration levels corresponded with yields, with intercropping treatments having lower capsaicin concentrations. In agreement with results reported by Álvarez-Solís et al. (2016) [37], no significant ($p < 0.05$) effect on capsaicin concentration was found due to intercropping jalapeño peppers with onion (*Allium cepa* L.), reporting values of 0.31 mg g$^{-1}$. Pinedo-Guerrero et al. (2017) [38] found capsaicin levels of 170 to 270 mg kg$^{-1}$ (equal to 0.17 to 0.27 mg g$^{-1}$). Jalapeño pepper capsaicin concentration levels, similar or higher that those reported in the present study were reported by Victoria-Campos et al. (2015) [39], who found capsaicin concentrations of 0.87 mg kg$^{-1}$ for fresh green jalapeño peppers, but values as high as 2.32 mg kg$^{-1}$ for grilled green peppers. The pepper fruit firmness in this study was very high, as compared with that reported by Cervantes-Paz et al. (2014) [40], who found a value of 43.9 N for brown peppers. When comparing the color attributes, it can be concluded that the peppers in our study looked attractive, since they were lighter (L*), greener (a*) and yellower (b*), than the color attributes reported in other papers. Mendoza-Sánchez et al. (2015) [41] reported color values for L*, a*, b*, Chroma and Hue of jalapeño peppers, after zero days of storage of 28.1, −8.9, 10.3, 12.9, and 46.1, respectively. Villarreal-Alba et al. (2004) [42] reported values ranging from 0.52 to 4.11 for L*, from −0.17 to −5.18 for a*, and from 0.61 to 4.42 for b*, across treatments of temperature and time of exposure to temperatures.

### 4.3. Microclimatic Variables

Such large PAR reductions in the intercropped treatments clearly affected photosynthesis and thus fruit yields and plant size characteristics. Díaz-Pérez (2013) [30] applied five shading treatment levels—0% (full sunlight), 30%, 47%, 62%, and 80%—representing about 2000, 1500, 1000, 750, and 300 μmol m$^{-2}$ s$^{-1}$, with net photosynthesis about 100%, 100%, 88%, 72%, and 48%, respectively. Thus, net photosynthesis was 100% at around 2000 and 1500 μmol m$^{-2}$ s$^{-1}$, 88% with 1000 μmol m$^{-2}$ s$^{-1}$, 72% with 750 μmol m$^{-2}$ s$^{-1}$, and only 48% with 300 μmol m$^{-2}$ s$^{-1}$. Furthermore, Gong et al. (2015) [14] suggested that photosynthesis in the shaded species in the intercrop system (soybeans) was light deficient when PAR was lower than 500 μmol m$^{-2}$ s$^{-1}$. Thus, since in this study the overall PAR mean value of the intercropped treatments was less than 300 μmol m$^{-2}$ s$^{-1}$, it can be concluded that photosynthesis was inhibited by a deficit of PAR, caused by corn shading to pepper plants in the

intercropped treatments. Ghanbari et al. (2010) [28] suggested that soil temperature, which is closely related with air temperature, depended on plant cover. They found that the highest soil temperature was observed in corn monocrop, intermediate in cowpea–corn intercropping, and the lowest in sole cowpea. Relative humidity measurements were collected on the same dates as temperatures (16 April and 12 May 2016) and found that plants grown with intercropping had about 20 and 30% higher relative humidity than plants without intercropping, respectively (Figure 4).

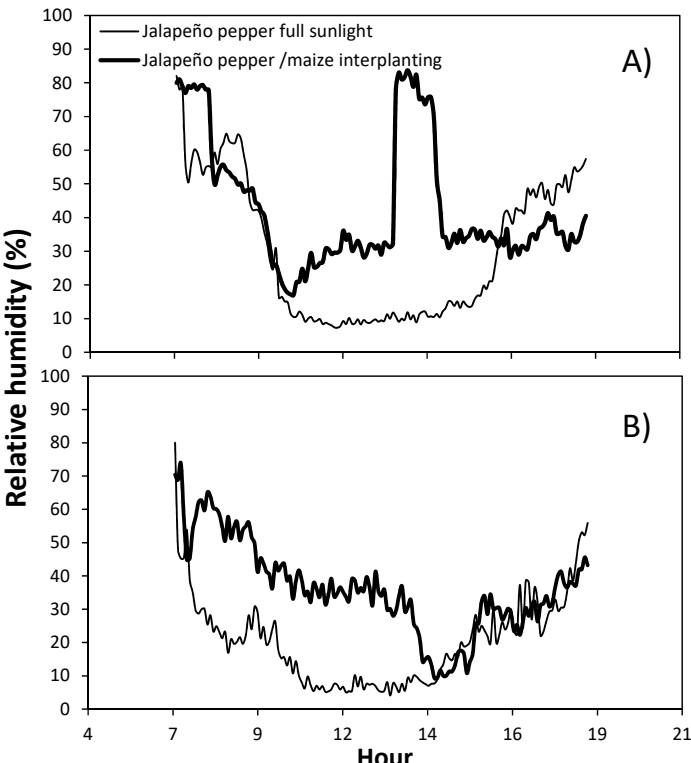

**Figure 4.** Relative humidity recorded at jalapeño pepper canopy level as affected by intercropping, on (**A**) 16 April and (**B**) 12 May 2016.

### 4.4. Pepper Plant Morphology

In our study, the PMu, a monocrop unshaded treatment, produced the tallest pepper plants, in contrast to several research reports which suggest that one of the morphological adaptations of shaded plants in intercropping systems is stem elongation. Díaz-Pérez (2013) [30] found a positive linear relation between plant height and shading level; plants growing without shading measured 90 cm, while 80% shading produced plants with an average of about 100 cm. Similar results for plant height were reported by Morano et al. (2017) [43], who also found that the stem thickness and plant leaf area showed opposite trends. Liu et al. (2017) [17] measured the morphological changes on soybeans when grown as monocrop or in two intercropping systems and found that soybean plant height in the two intercropping treatments was 51% higher than that in the monoculture. Similarly, Gong et al. (2015) [14] tested two soybean varieties grown as monocrop and soybeans–corn intercrop. They found that both varieties had longer, but slimmer main stems in shade-grown soybeans than in monocrop. A possible explanation for our shaded pepper plants not being taller than the non-shaded treatments is that shaded pepper plants were only gradually shaded as the corn grew over time, and not from the beginning of the jalapeños' growing season. Furthermore, it is worthwhile to bear in mind that stem elongation, if the result of a light deficit, can cause the phenomenon of etiolation, which is a physiological disorder that can lead to plant death and null or severely decreased productivity. Plants

in PMu treatment were not etiolated, but large and healthy, as demonstrated by consistently recording greater plant growth characteristics (LA, SLA, LDW, SDW, FDW, and TDW).

A possible explanation for the greater plant size attributes and crop yield of the PMu treatment may be due to both lack of competition and, possibly, the effect of plastic mulching in reducing soil salinity. Baath et al. (2017) [44] found that pepper plant heights significantly decreased with increasing irrigation water salinity 70 days after emergence. In research on small fruit pepper [45], the plant size increased up to 3.8–4.1 dS m$^{-1}$ electric conductivity (EC) of irrigation water, depending on the crop season. Also, soil salinity significantly reduced rapeseed (*Brassica napus* L.) plant height [46]. Similarly, Qados (2011) [47] found that salinity caused an increase in bean plant (*Vicia faba* L.) height with low and medium concentrations and a decrease with the highest concentration. The other mulched treatment, PMu+IC, possibly failed to grow and to yield as much as PMu due to the competition with corn. The BS treatment likely did not benefit from reduced soil salinity by plastic mulching, but had no competition with corn. The BS plants were relatively small, evidencing some form of stress, but gave no significantly different yields than those of the PMu treatment. The BS+IC treatment produced smaller pepper plants compared to the mulched sole crop, due to the higher salinity plus competence with corn. Additionally, this treatment experienced the high temperatures occurring toward the end of the two seasons.

## 5. Conclusions

In conclusion, far from protecting the pepper crop, corn is an aggressive competitor and therefore not suitable for intercropping in the already stressed environment of northwest Mexico. However, it may be worthwhile to test other crops grown in the region that may compete less aggressively with the main high-value crop. More research is needed to find sound strategies to reduce heath stress in vegetable crops grown in northwest Mexico, especially under a worsening of climatic conditions posed by climate change and soil salinity.

**Author Contributions:** Conceptualization, F.N.-R.; Data curation, I.E.-G. and V.C.-S.; Formal analysis, F.N.-R.; Funding acquisition, C.R.-A. and F.N.-R.; Investigation, A.G.-L. and F.N.-R.; Methodology, A.G.-L., I.E.-G. and V.C.-S.; Supervision, A.M.-M.; Writing—original draft, J.S.-C.; Writing—review & editing, J.S.-C.

**Funding:** This research received no external funding.

**Conflicts of Interest:** The authors declare no conflict of interest.

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
