# Peer review of "Assessment of Intercropping and Plastic Mulch as Tools to Manage Heat Stress, Productivity and Quality of Jalapeño Pepper"

_agronomy, doi:10.3390/agronomy8120307_

Round 1

Reviewer 1 Report

I am sending an annotated manuscript with comments and suggestions

Author Response

Mexicali, B. C., Mexico. December 3, 2019

Letter for Reviewer 2

Dear Reviewer:

In this new version of our paper we have rigorously revised the English language.

We did not find your annotated manuscript with comments and suggestions, but with the observations from the other two reviewers, the paper improved substantially.

We appreciate you revision.  

Kind Regards

Reviewer 2 Report

All corrections and comments are inserted in the attached file.

Author Response

Mexicali, B. C., Mexico. December 3, 2019

Letter for Reviewer 2

Dear Reviewer:

In the new version of our paper, we have corrected every one of your pertinent observations. The paper after these changes has substantially improved.

We appreciate you revision.  

Kind Regards

Jesús Santillano-Cázares, Ph. D.

Professor-Research, ICA-UABC

Reviewer 3 Report

All  my recommendations across the text and the tables should be addressed by the authors.

Author Response

Letter for Reviewer 3

Dear Reviewer:

Your review was the one that helped our paper the most. In the new version of our paper we have corrected every observation was made and included the suggested references, which strengthened our findings and discussion.

Regarding the number of Jalapeño peppers per unit of area, I have been explained by Dr. Fidel Núñez-Ramírez, my fellow Professor, who actually took care of the field experiments, that we had 3.12 plants per m2, since the beds measured 0.8 m wide, and plants were 0.4 m apart, thus in 1 m, a mean of 2.5 plants were planted, but there were 2 rows of plants per bed, so we had a mean of 5 plants per 0.8 m2, thus resulting in 3.1 plants per m2.

We appreciate your careful revision.  

Kind Regards

Jesús Santillano-Cázares, Ph. D.

Professor-Research, ICA-UABC

Round 2

Reviewer 3 Report

In my opinion, the manuscript in object can now be accepted for publication in Agronomy. However, some minor amendments relevant to text formatting should be performed.   

Author Response

Dear reviewer,

In this new version of our manuscript , all your observations were attended.

Besides correcting all the writing style and correctness of the minor English observations, Table 2 was removed; the micromol symbol was corrected since our previous submission. The calculation formula for growing degree days was included in Figure 3.

We hope this new version of our manuscript is now acceptable for publication.

Best Regards. 
